# ATOMIC COMPRESSION NETWORKS

## ABSTRACT

Compressed forms of deep neural networks are essential in deploying large-scale computational models on resource-constrained devices. Contrary to analogous domains where large-scale systems are build as a hierarchical repetition of small-scale units, the current practice in Machine Learning largely relies on models with non-repetitive components. In the spirit of molecular composition with repeating atoms, we advance the state-of-the-art in model compression by proposing *Atomic Compression Networks (ACNs)*, a novel architecture that is constructed by recursive repetition of a small set of neurons. In other words, the same neurons with the same weights are stochastically re-positioned in subsequent layers of the network. Empirical evidence suggests that ACNs achieve compression rates of up to three orders of magnitudes compared to fine-tuned fully-connected neural networks ($88\times$ to $1116\times$ reduction) with only a fractional deterioration of classification accuracy (0.15% to 5.33%). Moreover our method can yield sub-linear model complexities and permits learning deep ACNs with less parameters than a logistic regression with no decline in classification accuracy.

## 1 INTRODUCTION

The universe is composed of matter, a physical substance formed by the structural constellation of a plethora of unitary elements denoted as atoms. The type of an atom eventually defines the respective chemical elements, while structural bonding between atoms yields molecules (the building blocks of matter and our universe). In Machine Learning a neuron is the infinitesimal nucleus of intelligence (i.e. {atom, matter} ↔ {neuron, AI}), whose structural arrangement in layers produces complex intelligence models. Surprisingly, in contrast to physical matter where molecules often reuse quasi-identical atoms (i.e. repeating carbon, hydrogen, etc.), neural networks do not share the same neurons across layers. Instead, the neurons are parameterized through weights which are optimized independently for every neuron in every layer. Inspired by nature, we propose a new paradigm for constructing deep neural networks as a recursive repetition of a fixed set of neurons. Staying faithful to the analogy we name such models as *Atomic Compression Networks (ACNs)*. Extensive experimental results show that by repeating the same set of neurons, ACNs achieve unprecedented compression in terms of the total neural network parameters, with a minimal compromise on the prediction quality.

Deep neural networks (DNN) achieve state-of-the-art prediction performances on several domains like computer vision (Huang et al., 2018; Tan & Le, 2019) and natural language processing (Vaswani et al., 2017; Gehring et al., 2017). Therefore, considerable research efforts are invested in adopting DNNs for mobile, embedded, or Internet of Things (IoT) devices (Kim et al., 2015). Yet, multiple technical issues related to restricted resources, w.r.t. computation and memory, prevent their straightforward application in this particular domain (Han et al., 2016; Samie et al., 2016; Mehta et al., 2018). Even though prior works investigate neural compression techniques like pruning or low-rank parameter factorization, they face fragility concerns regarding the tuning of hyperparameters and network architecture, besides struggling to balance the trade-off between compression and accuracy (Cheng et al., 2017).

**Contributions:** In a nutshell, this paper introduces:

- a novel *compression paradigm for neural networks* composed of repeating neurons as the atomic network components and further motivated by function composition;

- compression rates of *up to three orders of magnitudes* compared to a cross-validated fully-connected network on nine real-world vector datasets;

- first work to achieve *sub-linear model complexities* measured in the number of trained parameters compared to connected architectures on several computer vision tasks.

## 2 RELATED WORK

### 2.1 MODULAR NEURAL NETWORKS

Our approach of training a set of neurons and (re-)using them in building the network architecture is partially related to the existing scheme of modular neural networks (MNN). End-to-End Module Networks (Andreas et al., 2016a;b; Hu et al., 2017) are deep neural network models that are constructed from manual module blueprints defined for different sub-tasks in question answering. The Compositional Recursive Learner proposed by Chang et al. (2018) employs a curriculum learning approach to learn modular transformations while Routing Networks (RN) (Rosenbaum et al., 2017) consist of a set of pre-defined modules (which each can be a NN) and a meta controller (called router) that decides in which order these modules are consecutively applied to a given input. Modular Networks (Kirsch et al., 2018) are an extension to RNs employing conditional computation with Expectation-Maximization. Finally the recent work of Cases et al. (2019) focuses on the recursive formulation of the Routing Network, consecutively applying one of a set of modules to an input. The crucial difference to our approach is that our neurons are much smaller than the general modules of MNN and that we reuse the same components on the same network level (e.g. within the same layer) while the modules of MNN are only applied sequentially. A different extension of the RN is the model of Zaremoodi et al. (2018) which uses the router in a gating mechanism to control the input to a set of shared RNN modules similar to a Mixture of Experts model (Jacobs et al., 1991). Although Mixture of Experts models are related, they normally do not stack and recursively reuse components within the same network but have a comparably shallow architecture.

Another related field of research is that of neural architecture search (NAS) (Zoph & Le, 2016). It is concerned with the automatic discovery of high performing neural network architectures for diverse tasks. There exists a multitude of search-approaches including (Neuro-)Evolution (Stanley et al., 2009; Real et al., 2017) and Reinforcement Learning (Zoph & Le, 2016; Pham et al., 2018). A sub-field of NAS is the dedicated search for neural modules called *cells* or *blocks* (Liu et al., 2018; Zoph et al., 2018; Pham et al., 2018), which can be reused in the network architecture. Although this is related to our approach, the discovered cells are usually much more complex than single neurons and only share their architecture while each cell has a different set of weights, therefore not reducing the total number of required parameters. Although parameter sharing approaches exist, they either share weights between different evolved macro-architectures (Real et al., 2017; Elsken et al., 2018) or only use them to warm start cells with a parameter copy on initialization, which then is refined in subsequent fine tuning steps (Pham et al., 2018; Xie et al., 2018), missing the advantage of the recursive sharing scheme employed by ACNs. Although recent works like Stamoulis et al. (2019) achieve huge speed-ups, the large amount of time required to find suitable architectures remains a major disadvantage, while our approach can be trained from scratch in one go.

### 2.2 NETWORK COMPRESSION

One popular way for network compression is the pruning of unimportant weights from the network, an idea first appearing several decades ago to prevent over-fitting by reducing the network complexity (LeCun et al., 1990). The seminal work by Han et al. (2015b) proposes an iterative procedure consisting of training and pruning weights with small magnitudes. Li et al. (2016) focus on pruning filters of trained CNNs based on the $L_1$-norm while Louizos et al. (2017b) and Li & Ji (2019) consider $L_0$-norm regularization. In contrast Luo et al. (2017) employ a pruning schedule on the channel with the smallest effect on the activations of the subsequent layer. Molchanov et al. (2016; 2019) use Taylor expansion to calculate importance scores for different channels to prune CNNs. Similarly neuron importance scores based on the reconstruction error of responses in the final layers are used by Yu et al. (2018) while Wang et al. (2018) propose an adaptive pruning scheme for sub-groups of weights. Furthermore there also exist recent Bayesian approaches to weight pruning (Molchanov et al., 2017; Louizos et al., 2017a; Ullrich et al., 2017).

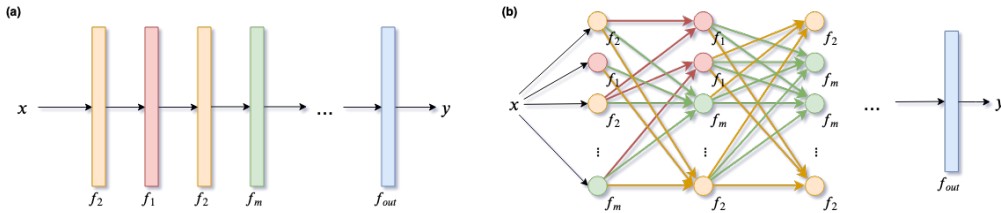

Figure 1: **(a)** LayerNet architecture; **(b)** ACN with single neuron components. The input masks are shown by colored arrows to respective neurons, where $f_1$ (red) has 2, $f_2$ (orange) has 3 and $f_m$ (green) has 4 inputs. The output layer $f_{out}$ (blue) is an independent module which is not reused.

Another path of research utilizes factorization approaches to produce low-rank approximations to the weight matrices of DNNs (Denton et al., 2014; Gong et al., 2014). Alternatively vector quantization methods can be applied to the parameter matrices, e.g. dividing the weights into sub-groups which are represented by their centroid (Denil et al., 2013; Gong et al., 2014). SqueezeNet introduced by Iandola et al. (2016) adapts so-called squeeze modules which reduce the number of channels while TensorNet (Novikov et al., 2015) uses the Tensor-Train format to tensorize the dense weight matrices of FC layers. Finally Wu (2016) use the Kronecker product to achieve layer-wise compression.

Recent work combines several of the aforementioned methods in hybrid models. Han et al. (2015a) use pruning together with layer-wise quantization and additional Huffman coding. In Kim et al. (2015) Tucker decomposition supported by a variational Bayesian matrix factorization is employed while Tung & Mori (2018) jointly apply pruning and quantization techniques in a single framework.

A special variant of compression methods for DNNs is the teacher-student approach also known as network distillation (Ba & Caruana, 2014; Hinton et al., 2015; Luo et al., 2016), which utilizes the relation of two models, a fully trained deep teacher model and a significantly smaller and more shallow student model to distill the knowledge into a much more compressed form. There exist different approaches to the design and training of the student model, e.g. utilizing special regularization (Sau & Balasubramanian, 2016) or quantization techniques (Polino et al., 2018).

A crucial advantage of ACNs is that we do not waste time and computational resources on training a large, over-parameterized architecture which is prone to over-fitting and usually needs a lot of training data only to compress it afterwards (e.g. by pruning or distillation). Instead we directly train smaller models which bootstrap their capacity by recursively-stacked neurons and therefore are more compact and can be trained efficiently with less data. Furthermore, the aforementioned approaches are often quite limited in achieving large compression without degrading the prediction performance.

HashedNets (Chen et al., 2015) use random weight sharing within the parameter matrix of each layer and introduce an additional hashing formulation of the forward pass and the gradient updates. Our model is different from these approaches since we share neurons which can have an arbitrary number of weights and don't restrict the sharing to the same layer, but enable sharing across layers, too.

As an additional complementary method to the aforementioned approaches one can train networks with reduced numerical precision of their parameters (Courbariaux et al., 2014; Gupta et al., 2015). This achieves further compression and experiments show that it leads only to a marginal degradation of model accuracy. Taking this idea to the extreme Courbariaux et al. (2016) constrain the values of all weights and activations to be either +1 or -1.

## 3 ATOMIC COMPRESSION NETWORKS

A neuron can be understood as an arbitrary function $f(x; \theta) : \mathbb{R}^D \to \mathbb{R}^1$ defined by its parameters $\theta \in \mathbb{R}^D$. The idea is to stochastically apply the same neuron with the same weights to the outputs of neurons from the previous layer. The different functions or components $f_i$ then form a global set $M$, $f_1, f_2, ..., f_m \in M$ which defines the set of parameters (indexed by the neurons) which is available to the model. Additionally we add the output layer $f_{out}$ which is a unique FC layer that is not reused anywhere else in the network (see figure 1).

The most naive approach to recursion is repeating whole layers with shared weights. Each module is represented by a FC layer mapping from a layer input dimension $D_{in}$ to the output dimension $D_{out}$. The architecture is depicted in figure 1a. More generally this approach can also be written in terms of a function composition (see equation 1). Sharing layers forces them to have equal input and output dimensions, resulting in rectangular-shaped neural architectures. For comparison with our neuron level ACN in the ablation study in the experiment section we reference this approach with random sampling of the sequential network composition by *LayerNet*.

$$g(x) = f_m \circ f_{m-1} \circ ... \circ f_1(x), \quad f_1, f_2, ..., f_m \in M \tag{1}$$

In contrast to repeating entire layers, *ACN* reuses the most atomic components of a neural network which are represented by its single neurons. Consequently a full network layer in the model consists of several neurons which do not need to be all different, but the same neuron can be reused more than once in the same layer (see figure 1b). Each neuron $p \in M$ has an input dimension $d_p$ which we choose randomly from a preset range of dimensions $D$. When inserting the neuron into a layer $\ell$, we randomly sample $d_p$ connections to the neurons of the previous layer forming a mask $\delta_p^{(\ell)}$ on its activation $z$. While the trainable parameters (weights and bias) of the neurons are shared, this connection mask is different for every neuron in every layer. The procedure for sampling the ACN architecture is shown in algorithm 1.

---

**Algorithm 1:** ACN Architecture Sampling

    **input**  : $D$, number of neurons $N$, number of layers $L$, layer sizes $K \in \mathbb{N}^L$
    **output**: ACN architecture $m, \delta$

    // Create neurons with random dimensions
1  $M := \{f_1(\cdot; \theta_1), ..., f_N(\cdot; \theta_N)\}, \quad \theta_i \in \mathbb{R}^{d_i}, \; d_i \sim \mathcal{U}(1, D)$

    // Create network architecture
2  **for** $\ell = 1, ..., L$ **do**
3      $M^{(\ell)} := \{f_i(\cdot; \theta_i) \mid f_i(\cdot; \theta_i) \in M \wedge d_i < K^{(\ell-1)}\}$
4      **for** $i = 1, ..., K^{(\ell)}$ **do**
5         $f_j(\cdot; \theta_j) \sim M^{(\ell)}$
6         $m_i^{(\ell)}(\cdot) := f_j(\cdot; \theta_j)$
7         $\delta_i^{(\ell)} \sim \mathcal{U}\left(1, K^{(\ell-1)}\right)_{d_j}$
8  **return** $m, \delta$

---

Equation 2 shows the forward pass through a ACN neuron layer, where $\Pi$ represents the projection (selection) of the mask indices $\delta$ from the activation of the previous layers:

$$z_i^{(\ell)} = m_i^{(\ell)}\left(\Pi_{\delta_i^{(\ell)}}\left(z^{(\ell-1)}\right)\right), \quad i = 1, ..., K^{(\ell)} \tag{2}$$

To reduce the additional memory complexity of these input masks we can store the seed of the pseudo random generator used for sampling the masks and are able to reproduce them on the fly in the forward pass at inference time. Since the input masks of the neurons are completely random, situations may arise in which not all elements of $z$ are forwarded to the next layer. However in our experiments we find that this case happens only very rarely granted that the number of neurons per layer and the greater part of the input dimensions $D$ are sufficiently large.

## 4 EXPERIMENTS

### 4.1 PRELIMINARY STUDY OF MODULE RECURSION

To gain more insights into the potential of ACN we perform a small study on a synthetic curve fitting problem. We compare a simple FC network with a varying number of layers and neurons to an ACN with approximately the same number of parameters. The curve fitting problem is a regression task on the function $f(x) = 3.5x^3 - 3\sin(x) + 0.5\sin(8x^2 + 0.5\pi)^2$. To achieve the best possible fit for the models we perform a hyper-parameter grid search including the dimension of hidden layers, learning

rate and the number of different modules for the recursive net. Details can be found in appendix A.2. Considering our knowledge about the true function which includes powers of $x$ we boost the performance of both models by squaring the neuron activations.

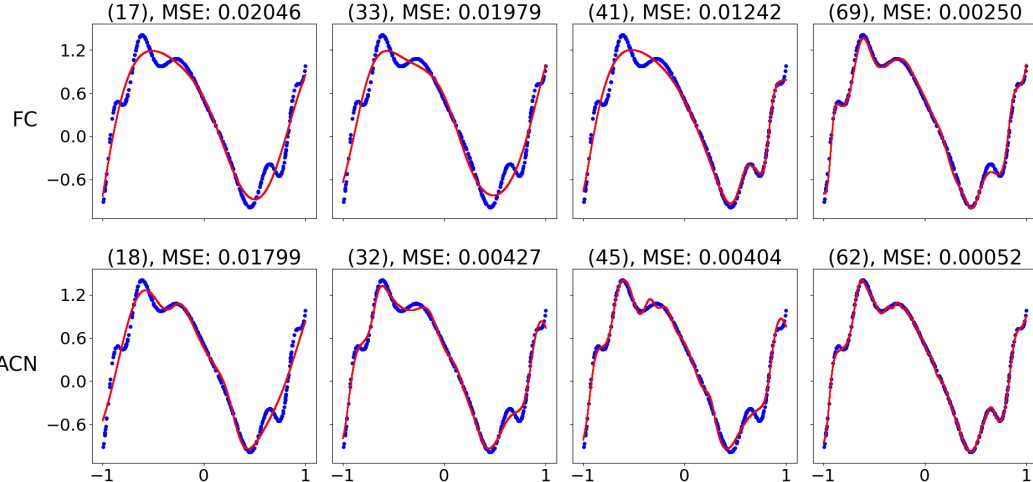

Figure 2: Plots of the model fit to the curve data. The first row shows the fit of the FC baseline, the second that of ACN. In the columns the respective number of model parameters as well as the achieved MSE on the test set are given above the plots. Since an arbitrary number of parameters cannot always be achieved with each model, the next nearest number with a competitive model was selected, e.g. in the first column with 17 and 18 parameters respectively.

As can be seen in figure 2, ACN consistently achieves a better fit in terms of MSE than the FC baseline. Already an ACN with only 32 parameters is sufficient to approximately fit even the highly non-linear parts of the function while the FC network only achieves a comparable fit with actually more than twice the number of parameters. This result can be explained by some intuition about function composition. Consider a function $f(x) = (\alpha x + \beta)^2, f : \mathbb{R} \to \mathbb{R}$ with sole parameters $\alpha$ and $\beta$. Then we can create much more complex functions with the same set of parameters by composition $f(f(x)) = \left(\alpha^3 x^2 + 2\alpha^2 \beta x + \alpha \beta^2 + \beta\right)^2$. Extending this to other functions $g, h, ...$ (which each can be a neuron) enables us to create complex functions with a small set of parameters. Furthermore compared to a standard FCN our ACN achieves much deeper architectures with the same number of parameters what further improves the fitting capability.

## 4.2 EXPERIMENTAL SETUP

We evaluate the performance of the proposed algorithms using nine publicly available real-world classification datasets and three image datasets. The selected datasets and their characteristics are detailed in appendix A.1. The datasets were chosen from the OpenML-CC18 benchmark[1] which is a curated high quality version of the OpenML100 benchmark (Bischl et al., 2017). Details on the employed hyper-parameters and training setup can be found in appendix A.2.

We compare our model to six different baseline models:

- **FC**: A standard FC network of comparable size. It was shown in Chen et al. (2015) that this is a simple but strong baseline which often outperforms more sophisticated methods.
- **RER**: Random Edge Removal first introduced in Cireşan et al. (2011). For this baseline a number of connections is randomly dropped from a FC network on initialization. In contrast to dropout (Srivastava et al., 2014) the selected connections are dropped completely and accordingly the network is trained and also evaluated as sparse version.
- **TensorNet** (Novikov et al., 2015): This model builds on a generalized low-rank quantization method that treats the weight matrices of FC layers as a two dimensional tensor which

---

[1]https://docs.openml.org/benchmark/#openml-cc18

can be decomposed into the Tensor-Train (TT) format (Oseledets, 2011). The authors also compare their model to HashedNet (Chen et al., 2015) and claim to outperform it by 1.2%. To train TensorNet on our datasets we introduce two FC layers mapping from the input to the consecutive TT layers and from the last TT layer back to the output dimension.

- **BC**: The Bayesian compression method of Louizos et al. (2017a) uses scale mixtures of normals as sparsity-inducing priors on groups of weights, extending the variational dropout approach of Molchanov et al. (2017) to completely prune unimportant neurons while training a model from scratch. We achieve varying levels of compression and accuracy by specifying different thresholds for the variational dropout rate of the model.

- **TP**: Molchanov et al. (2019) use first and second order Taylor expansions to compute importance scores for filters and weights which then are used to prune respective parts of the network. In our experiments we use the method which the authors report as best, introducing pruning gates directly after batch normalization layers. To realize different levels of compression we prune increasingly bigger parts of the network based on the calculated importance scores in an iterative fine tuning procedure.

- **LogisticRegression**: A linear model.

## 4.3 EXPERIMENTAL RESULTS

The results of the experimental study on the real-world datsets are shown in figure 3 while table 1 shows the results for the image datasets. All results are obtained by running each model-hyper-parameter combination with three different random seeds and averaging the resulting performance metric and number of parameters. The standard deviation between runs can be found in table 5 in appendix A.3.

Table 1: Classification test accuracy on image datasets — in each row the best model (selected based on validation accuracy) up to the specified maximum number of parameters is shown. Therefore the model and its test accuracy values do not change for greater numbers of parameters, if a smaller model has a better validation accuracy. For some models very small versions could not be trivially achieved (indicated by "-"). All trained models are fully connected architectures. We did not train ACN for more than 500,000 parameters.

| | # Parameters | FC | RER | TensorNet | BC | TP | *LayerNet* | ACN |
|---|---|---|---|---|---|---|---|---|
| **MNIST** | < 5,000 | 0.9265 | 0.5114 | - | 0.3758 | 0.1434 | - | **0.9445** |
| | < 10,000 | 0.9350 | 0.5114 | - | 0.8772 | 0.2482 | 0.9433 | **0.9625** |
| | < 50,000 | 0.9650 | 0.7508 | - | 0.9763 | 0.8249 | 0.9711 | **0.9774** |
| | < 100,000 | 0.9650 | 0.8806 | 0.9532 | 0.9764 | 0.9236 | 0.9711 | **0.9774** |
| | < 500,000 | 0.9844 | 0.9440 | 0.9619 | 0.9805 | 0.9709 | **0.9847** | 0.9774 |
| | ≥ 500,000 | 0.9852 | 0.9440 | 0.9619 | 0.9805 | 0.9737 | **0.9854** | - |
| **FMNIST** | < 5,000 | - | 0.6055 | - | 0.4765 | 0.2704 | - | **0.8513** |
| | < 10,000 | 0.8489 | 0.6055 | - | 0.8220 | 0.2714 | 0.8478 | **0.8669** |
| | < 50,000 | 0.8672 | 0.7514 | - | 0.8713 | 0.6750 | 0.8684 | **0.8795** |
| | < 100,000 | 0.8672 | 0.8137 | 0.8568 | 0.8713 | 0.8273 | 0.8684 | **0.8795** |
| | < 500,000 | 0.8960 | 0.8459 | 0.8676 | 0.8741 | 0.8612 | **0.8965** | 0.8795 |
| | ≥ 500,000 | **0.8985** | 0.8459 | 0.8676 | 0.8741 | 0.8635 | 0.8963 | - |
| **CIFAR10** | < 5,000 | - | - | - | - | 0.0999 | - | **0.4557** |
| | < 10,000 | 0.3932 | - | - | 0.2577 | 0.1508 | - | **0.4795** |
| | < 50,000 | 0.4183 | 0.3039 | - | 0.5015 | 0.2167 | 0.4195 | **0.5138** |
| | < 100,000 | 0.4745 | 0.3039 | - | 0.5015 | 0.3144 | 0.4195 | **0.5138** |
| | < 500,000 | 0.5343 | 0.4055 | 0.4750 | 0.5015 | 0.4133 | **0.5364** | 0.5138 |
| | ≥ 500,000 | **0.5567** | 0.4293 | 0.4893 | 0.5141 | 0.4970 | 0.5559 | - |

As can be seen in figure 3, ACN consistently outperforms all other models on all of the nine real-world datasets for very small models, for datasets with a huge feature space even for models with up to 10,000 parameters. This shows that the ACN architecture is very efficient w.r.t. its parameters and the required amount of training-data and that on these datasets the recursive nature and parameter sharing throughout the architecture has a positive impact on the predictive efficiency compared to the number of parameters. For larger models the advantage of ACN decreases and the other models catch

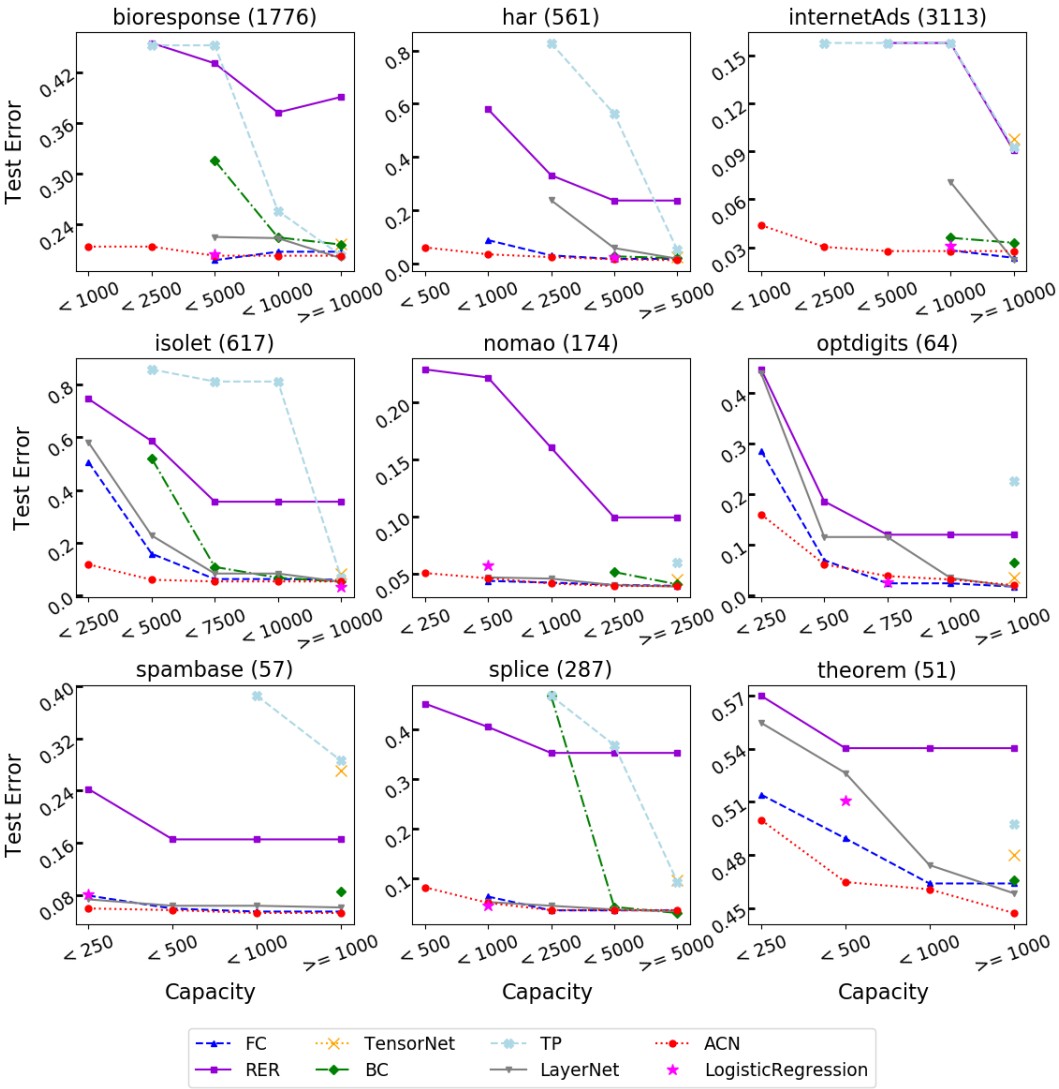

Figure 3: Plots of the model classification error on the test set for different numbers of parameters. For each tick on the x-axis the model with the best validation accuracy under the threshold for the number of parameters is selected. In the titles next to the name of the dataset we indicate the dimension of the respective feature space. For some methods very small models could not be achieved so they have no points plotted on the left part of the plots (e.g. a fully connected network will not have less parameters than a linear model with $number\_of\_inputs \times number\_of\_classes$ many parameters)

up or even outperform it by a small margin, e.g. in case of the *InternetAds* dataset, however ACN remains competitive. TensorNet usually is at least a factor of two larger than the other models and therefore is only shown in the last bin. The same applies for TP and BC on the optdigits, theorem and nomao/spambase datasets respectively.

To evaluate the parameter efficiency of our technique we compare ACN and FC (as the best performing baseline) to a large optimal FC network trained on the same dataset (doing a hyper-parameter search with the same parameters mentioned in section 4.2 and five increasingly large architectures). We show the results regarding the relative compression rate and the respective loss in model performance in table 2. Say $n$ is the number of model parameters of the large baseline and $n'$ that of the evaluated model, then the compression rate is given by $\rho = n/n'$. Our model achieves compression rates of 88 up to more than 1100 times while the loss in test accuracy is kept at around 0.15% up to 5.33%. In comparison FC achieves compression of 67 up to 528 times with accuracy loss between 0.28% and

7.13%. For all datasets except *spambase* and *theorem* ACNs realize higher compression rates than FC as well as a smaller loss in accuracy in 5/9 cases. However since the comparison to a large model baseline can be biased by model selection and the thoroughness of the respective hyper-parameter search, we also investigate how the performance and capacity of the models compare w.r.t. a linear baseline. The results in table 2 confirm the trend that is observed in the comparison to the large model.

Table 2: Compression and accuracy change compared to a large FC model and a linear model baseline

| | Compared to large FC | | | | Compared to Linear Model | | | |
| | small FC | | small ACN | | small FC | | small ACN | |
| Dataset | $\rho$ | $\Delta_{Acc}(\%)$ | $\rho$ | $\Delta_{Acc}(\%)$ | $\rho$ | $\Delta_{Acc}(\%)$ | $\rho$ | $\Delta_{Acc}(\%)$ |
|---|---|---|---|---|---|---|---|---|
| 1) bioresponse | 67.41 | **-0.28** | **322.15** | -1.18 | 0.998 | **+0.70** | **4.771** | -0.90 |
| 2) HAR | 155.59 | -7.13 | **361.82** | **-4.40** | 2.942 | -6.25 | **6.840** | **-3.50** |
| 3) InternetAds | 133.23 | **-1.09** | **1115.58** | -2.66 | 0.998 | **+0.31** | **8.360** | -1.26 |
| 4) isolet | 74.49 | -3.71 | **87.76** | **-3.45** | 3.042 | -2.86 | **3.58** | **-2.60** |
| 5) nomao | 218.67 | **-0.74** | **460.36** | -1.43 | 0.972 | **+1.37** | **2.047** | +0.68 |
| 6) optdigits | 149.12 | -6.15 | **168.06** | **-5.33** | 1.879 | -4.41 | **2.117** | **-3.59** |
| 7) spambase | **129.59** | -2.12 | 92.46 | **-0.15** | **0.951** | +0,23 | 0.678 | **+2.20** |
| 8) splice | 211.46 | **-3.47** | **316.12** | -5.33 | 1.467 | **-1.73** | **2.193** | -3.59 |
| 9) theorem | **97.47** | -6.50 | 95.30 | **-4.03** | **0.647** | +2.14 | 0.633 | **+4.61** |

On the image datasets ACN consistently and significantly outperforms all other models for up to 100,000 parameters. On *MNIST* our smallest model has 4091 parameters and outperforms a linear model with 7850 by 1.8% although the baseline has nearly twice as many parameters. The same trend can be observed for *FashionMNIST* (7057 / 7850 with a difference of 2.04%) and *CIFAR10* where an ACN with 7057 parameters employs less than a quarter of the 30730 parameters needed by the simplest linear model and outperfoms it by a overwhelming 8.63%. Surprisingly the LayerNet outperfoms all other models for the category $< 500,000$. For more than 500,000 parameters the FC network performs best. Although the results on *CIFAR10* first might seem rather poor compared to the approximately 96% accuracy which is achieved by CNN (Zagoruyko & Komodakis, 2016), the best results achieved by dedicated FC networks lie around 56% and are only topped by a specifically designed network with linear bottleneck layers and unsupervised pre-training (Lin et al., 2015). While our model sacrifices some processing time for higher compression, the forward pass at inference time still works in a matter of milliseconds on a standard CPU.

## 4.4 Ablation Study on Neuron-Recursion

To investigate the effect of the chosen level of sharing and recursion we perform a small ablation study. We compare ACN with parameter-sharing and recursion on a neuron-level with the simpler approach of sharing and reusing whole layers described as the *LayerNet* architecture in section 3. Therefore we also report the performance for the LayerNet in an extra column in the tables 5 and 1. The results imply that in general the recursion on neuron-level is much more effective to achieve competitive models with high compression, outperforming the LayerNet in all but one case for less than 10,000 parameters. However the LayerNet seems to have a beneficial regularization effect on large models, what is notable especially on the image datasets.

## 5 Conclusion

In this paper we presented *Atomic Compression Networks (ACN)*, a new network architecture which recursively reuses neurons throughout the model. We evaluate our model on nine vector and three image datasets where we achieve promising results regarding the compression rate and the loss in model accuracy. In general ACNs achieve much tinier models with only a small to moderate decrease of accuracy compared to six other baselines. For future work we plan to include skip connections in the architecture and to extend the idea to CNNs and the sharing of kernel parameters as well as for the FC layers. Another interesting path of research is the combination of the ACN scheme with NAS methods to further optimize the efficiency and performance of the created architectures.

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

## A  APPENDIX

### A.1  DATASET DETAILS

Table 3 shows detailed attributes for the datasets selected from the OpenML-CC18 benchmark. The selection criteria were datasets with more than 3000 instances and more than 50 original features. The number of features reported in the table is the number of features after one-hot-encoding. For the image datasets (10)-(12) we employ the original train/test splits and use a random 20% of the training set for validation leading to a train/val/test split of 0.64/0.16/0.2. The datasets (1)-(9) are split randomly using the same fractions. We do not use any kind of additional transformation or data augmentation beyond those which was already originally applied to the respective datasets.

Table 3: Description of datasets

| Dataset | # Instances | # Features | # Classes | % Minor. Cl. |
|---|---|---|---|---|
| (1) bioresponse (Kaggle, 2012) | 3751 | 1776 | 2 | 45.77 |
| (2) HAR (Anguita et al., 2013) | 10299 | 561 | 6 | 13.65 |
| (3) InternetAds (Kushmerick, 1999) | 3279 | 3113 | 2 | 14.00 |
| (4) isolet (Fanty & Cole, 1991) | 7797 | 617 | 26 | 3.82 |
| (5) nomao (Candillier & Lemaire, 2012) | 34465 | 174 | 2 | 28.56 |
| (6) optdigits (Dua & Graff, 2017) | 5620 | 64 | 10 | 9.86 |
| (7) spambase (Dua & Graff, 2017) | 4601 | 57 | 2 | 39.40 |
| (8) splice (Noordewier et al., 1991) | 3190 | 287 | 3 | 24.04 |
| (9) theorem (Bridge et al., 2014) | 6118 | 51 | 6 | 7.94 |
| (10) MNIST (LeCun et al., 1998) | 70000 | 784 | 10 | 9.02 |
| (11) FashionMNIST (Xiao et al., 2017) | 70000 | 784 | 10 | 10.00 |
| (12) CIFAR10 (Krizhevsky et al., 2009) | 60000 | 3072 | 10 | 10.00 |

## A.2    Hyper-Parameter Configuration

Table 4: Hyper-Parameters for Preliminary Study (4.1)

| Hyper-Parameter | SearchGrid FC | SearchGrid ACN |
|---|---|---|
| hidden layer size | 2, 3, 4, 5, 8 | 3, 4, 5, 8 |
| learning rate | 0.1, 0.05, 0.01 | 0.1, 0.01 |
| number of layers | 1, 2, 3 | 2, 3, 4 |
| max gradient norm | 0.99, 1, 2, 5 | 0.99, 1, 2 |
| use batch norm | True, False | True, False |
| number of modules | - | 2, 4, 8 |

### Hyper-Parameter and Training Setup for real-world datasets

Given the large computational demands of deep learning architectures, a proper full hyper-parameter search is infeasible for the models employed on the real-world datasets. Under these circumstances we follow the established trend of using some specific deep architectures designed based on expert knowledge together with a small selected parameter grid containing learning rate (0.01, 0.001), use of batch normalization (Ioffe & Szegedy, 2015) and dropout probability (0.0, 0.3). To enable a fair comparison the hyper-parameter search for every model type which is trained from scratch is done for the same number of combinations. Accordingly for each dataset we evaluate eight different architectures per model type where we vary the number of hidden layers and the width of these layers, for ACN $M$ and $D$ are different between some of the eight varying architectures. The size of the set of modules $M$ is varied in the range between 3 and 5 for the LayerNet (ablation study) and between 16 and 512 for the ACN. The range of input dimensions $D$ is set to a subset of (2, 4, 8, 16, 32, 64, 128) depending on the maximum width of the used hidden layers and the respective input dimension. In case of the TensorNet we vary the dimension of the tensor-train layers and the length of the tensor-train decomposition between different architectures per dataset together with the aforementioned parameter grid. All other parameters remain at their default values (Novikov et al., 2015). For the two pruning approaches BC and TP we train two different large FC architectures with the described hyper-parameter grid and then prune the networks in small steps, selecting the best architecture within every parameter bin. The models are trained for 25 epochs on batches of size 64 for the real-world datasets and 128 for the image datasets. In order to minimize the loss we apply stochastic gradient descent update steps using Adam (Kingma & Ba, 2014) and halve the initial learning rate every 10 epochs. The only exception is Tp where we use SGD with a momentum term of 0.99 as proposed by the authors but employ the same learning rate schedule. Finally, in order to avoid exploding gradients the gradient norm is truncated to a maximum of 10. Apart from the TensorNet for which we adapt the publicly available Tensorflow Code[2], all models are implemented using the PyTorch 1.1 framework. For BC[3] and TP[4] we adapt the public PyTorch code base provided by the authors.

## A.3    Additional Results

In this section we show the results presented in figure 3 in tabular form in table 5. Furthermore for completeness we present additional results regarding two simple sparsification baselines in table 6 employing L1-regularization (**L1**) and iterative hard tresholding (**L1+HT**) (Han et al., 2015b) but without explicit cardinality constraint (Jin et al., 2016). The additional baseline models perform mostly on par with the small FC baseline, sometimes slightly better or slightly worst. The biggest change can be seen on the splice dataset where the additional baselines perform better than FC and ACN in the three bins with the highest number of parameters. The additional baselines were trained and evaluated following the same experimentation and hyperparameter protocol as the other models.

---

[2]https://github.com/timgaripov/TensorNet-TF

[3]https://github.com/KarenUllrich/Tutorial_BayesianCompressionForDL

[4]https://github.com/NVlabs/Taylor_pruning

Table 5: Test accuracy and its standard deviation on real-world datasets. The standard deviation of runs with FC, RER, BC and TP models are always below the displayed three decimals precision and therefore are not shown in the table.

| Capacity | FC | RER | TensorNet | BC | TP | *LayerNet* | ACN |
|---|---|---|---|---|---|---|---|
| *(1) bioresponse* | | | | | | | |
| < 1000 | - | - | - | - | - | - | **0.787** (±0.005) |
| < 2500 | - | 0.546 | - | - | 0.548 | - | **0.787** (±0.005) |
| < 5000 | **0.803** | 0.570 | - | 0.685 | 0.548 | 0.775 (±0.008) | 0.797 (±0.005) |
| < 10000 | 0.793 | 0.628 | - | 0.776 | 0.744 | 0.777 (±0.003) | **0.797** (±0.005) |
| ≥ 10000 | 0.793 | 0.609 | 0.784 (±0.013) | 0.784 | 0.797 | **0.800** (±0.004) | 0.797 (±0.005) |
| *(2) HAR* | | | | | | | |
| < 500 | - | - | - | - | - | - | **0.940** (±0.001) |
| < 1000 | 0.913 | 0.420 | - | - | - | - | **0.967** (±0.003) |
| < 2500 | 0.971 | 0.669 | - | - | 0.171 | 0.763 (±0.108) | **0.977** (±0.005) |
| < 5000 | **0.984** | 0.764 | - | 0.974 | 0.438 | 0.943 (±0.044) | **0.985** (±0.004) |
| ≥ 5000 | 0.983 | 0.764 | 0.975 (±0.001) | 0.981 | 0.950 | 0.983 (±0.002) | **0.989** (±0.002) |
| *(3) InternetAds* | | | | | | | |
| < 1000 | - | - | - | - | - | - | **0.956** (±0.012) |
| < 2500 | - | - | - | - | 0.842 | - | **0.970** (±0.002) |
| < 5000 | - | 0.842 | - | - | 0.842 | - | **0.972** (±0.005) |
| < 10000 | **0.972** | 0.842 | - | 0.964 | 0.842 | 0.929 (±0.062) | **0.972** (±0.005) |
| ≥ 10000 | **0.977** | 0.909 | 0.902 (±0.002) | 0.967 | 0.908 | **0.978** (±0.004) | 0.972 (±0.005) |
| *(4) isolet* | | | | | | | |
| < 2500 | 0.494 | 0.254 | - | - | - | 0.420 (±0.124) | **0.883** (±0.017) |
| < 5000 | 0.842 | 0.414 | - | 0.480 | 0.141 | 0.773 (±0.030) | **0.941** (±0.001) |
| < 7500 | 0.939 | 0.645 | - | 0.893 | 0.188 | 0.918 (±0.009) | **0.948** (±0.002) |
| < 10000 | 0.939 | 0.645 | - | 0.934 | 0.188 | 0.918 (±0.009) | **0.948** (±0.002) |
| ≥ 10000 | 0.941 | 0.645 | 0.917 (±0.006) | **0.949** | 0.932 | **0.949** (±0.002) | **0.948** (±0.002) |
| *(5) nomao* | | | | | | | |
| < 250 | - | 0.771 | - | - | - | - | **0.950** (±0.003) |
| < 500 | **0.957** | 0.778 | - | - | - | 0.953 (±0.001) | 0.954 (±0.001) |
| < 1000 | **0.958** | 0.840 | - | - | - | 0.955 (±0.001) | **0.959** (±0.001) |
| < 2500 | **0.960** | 0.901 | - | 0.949 | - | **0.960** (±0.001) | **0.961** (±0.003) |
| ≥ 2500 | **0.961** | 0.901 | 0.955 (±0.001) | 0.960 | 0.940 | **0.962** (±0.001) | **0.961** (±0.003) |
| *(6) optdigits* | | | | | | | |
| < 250 | 0.714 | 0.552 | - | - | - | 0.560 (±0.118) | **0.840** (±0.012) |
| < 500 | 0.931 | 0.814 | - | - | - | 0.885 (±0.006) | **0.939** (±0.011) |
| < 750 | **0.977** | 0.881 | - | - | - | 0.885 (±0.006) | 0.963 (±0.004) |
| < 1000 | **0.977** | 0.881 | - | - | - | 0.966 (±0.005) | 0.969 (±0.002) |
| ≥ 1000 | **0.984** | 0.881 | 0.965 (±0.004) | 0.936 | 0.775 | 0.984 (±0.003) | 0.980 (±0.004) |
| *(7) spambase* | | | | | | | |
| < 250 | 0.921 | 0.758 | - | - | - | 0.927 (±0.012) | **0.941** (±0.003) |
| < 500 | 0.941 | 0.835 | - | - | - | 0.936 (±0.004) | **0.943** (±0.008) |
| < 1000 | 0.945 | 0.835 | - | - | 0.615 | 0.936 (±0.004) | **0.948** (±0.004) |
| ≥ 1000 | 0.945 | 0.835 | 0.729 (±0.006) | 0.915 | 0.714 | 0.939 (±0.007) | **0.948** (±0.004) |
| *(8) splice* | | | | | | | |
| < 500 | - | 0.549 | - | - | - | - | **0.917** (±0.025) |
| < 1000 | 0.936 | 0.596 | - | - | - | 0.947 (±0.013) | **0.949** (±0.006) |
| < 2500 | **0.964** | 0.648 | - | 0.533 | 0.533 | 0.954 (±0.006) | **0.963** (±0.013) |
| < 5000 | **0.964** | 0.648 | - | 0.957 | 0.632 | 0.962 (±0.004) | **0.963** (±0.013) |
| ≥ 5000 | 0.964 | 0.648 | 0.902 (±0.036) | **0.969** | 0.906 | 0.965 (±0.003) | 0.963 (±0.013) |
| *(9) theorem* | | | | | | | |
| < 250 | 0.486 | 0.430 | - | - | - | 0.445 (±0.006) | **0.500** (±0.013) |
| < 500 | 0.511 | 0.460 | - | - | - | 0.474 (±0.026) | **0.535** (±0.007) |
| < 1000 | 0.536 | 0.460 | - | - | - | 0.526 (±0.004) | **0.540** (±0.007) |
| ≥ 1000 | 0.536 | 0.460 | 0.520 (±0.003) | 0.535 | 0.503 | 0.542 (±0.004) | **0.553** (±0.013) |

Table 6: Test accuracy and its standard deviation on real-world datasets for the additional sparsification baselines. The standard deviation of runs with FC, L1 and L1+HT models are always below the displayed three decimals precision and therefore are not shown in the table.

| Capacity | FC | L1 | L1+HT | ACN |
|---|---|---|---|---|
| *(1) bioresponse* | | | | |
| < 1000 | - | 0.548 | 0.548 | **0.787** (±0.005) |
| < 2500 | - | 0.548 | 0.777 | **0.787** (±0.005) |
| < 5000 | **0.803** | 0.767 | 0.791 | 0.797 (±0.005) |
| < 10000 | 0.793 | 0.788 | 0.791 | **0.797** (±0.005) |
| ≥ 10000 | 0.793 | 0.788 | 0.791 | 0.797 (±0.005) |
| *(2) HAR* | | | | |
| < 500 | - | - | - | **0.940** (±0.001) |
| < 1000 | 0.913 | 0.181 | 0.193 | **0.967** (±0.003) |
| < 2500 | 0.971 | 0.181 | 0.959 | **0.977** (±0.005) |
| < 5000 | **0.984** | 0.981 | 0.967 | **0.985** (±0.004) |
| ≥ 5000 | 0.983 | 0.981 | 0.975 | **0.989** (±0.002) |
| *(3) InternetAds* | | | | |
| < 1000 | - | 0.916 | 0.913 | **0.956** (±0.012) |
| < 2500 | - | 0.916 | 0.966 | **0.970** (±0.002) |
| < 5000 | - | 0.969 | 0.966 | **0.972** (±0.005) |
| < 10000 | 0.972 | **0.977** | 0.966 | 0.972 (±0.005) |
| ≥ 10000 | **0.977** | **0.977** | 0.966 | 0.972 (±0.005) |
| *(4) isolet* | | | | |
| < 2500 | 0.494 | 0.033 | 0.488 | **0.883** (±0.017) |
| < 5000 | 0.842 | 0.033 | 0.933 | **0.941** (±0.001) |
| < 7500 | 0.939 | 0.938 | 0.933 | **0.948** (±0.002) |
| < 10000 | 0.939 | 0.938 | **0.953** | 0.948 (±0.002) |
| ≥ 10000 | 0.941 | 0.938 | **0.953** | 0.948 (±0.002) |
| *(5) nomao* | | | | |
| < 250 | - | - | - | **0.950** (±0.003) |
| < 500 | **0.957** | - | - | 0.954 (±0.001) |
| < 1000 | 0.958 | 0.718 | 0.718 | **0.959** (±0.001) |
| < 2500 | 0.960 | 0.952 | 0.949 | **0.961** (±0.003) |
| ≥ 2500 | **0.961** | 0.952 | 0.951 | **0.961** (±0.003) |
| *(6) optdigits* | | | | |
| < 250 | 0.714 | - | - | **0.840** (±0.012) |
| < 500 | 0.931 | - | - | **0.939** (±0.011) |
| < 750 | **0.977** | - | 0.104 | 0.963 (±0.004) |
| < 1000 | **0.977** | 0.104 | 0.961 | 0.969 (±0.002) |
| ≥ 1000 | **0.984** | 0.979 | **0.985** | 0.980 (±0.004) |
| *(7) spambase* | | | | |
| < 250 | 0.921 | - | - | **0.941** (±0.003) |
| < 500 | 0.941 | - | - | **0.943** (±0.008) |
| < 1000 | 0.945 | 0.733 | 0.739 | **0.948** (±0.004) |
| ≥ 1000 | 0.945 | 0.919 | 0.922 | **0.948** (±0.004) |
| *(8) splice* | | | | |
| < 500 | - | - | - | **0.917** (±0.025) |
| < 1000 | 0.936 | 0.533 | 0.533 | **0.949** (±0.006) |
| < 2500 | 0.964 | 0.889 | **0.974** | 0.963 (±0.013) |
| < 5000 | 0.964 | 0.971 | **0.974** | 0.963 (±0.013) |
| ≥ 5000 | 0.964 | **0.976** | **0.977** | 0.963 (±0.013) |
| *(9) theorem* | | | | |
| < 250 | 0.486 | - | - | **0.500** (±0.013) |
| < 500 | 0.511 | - | - | **0.535** (±0.007) |
| < 1000 | 0.536 | 0.422 | 0.422 | **0.540** (±0.007) |
| ≥ 1000 | 0.536 | 0.488 | 0.493 | **0.553** (±0.013) |

