# OpenReview forum: "Atomic Compression Networks"
_ICLR.cc/2020/Conference — Reject_

### Official Review · AnonReviewer2 · 2019-10-18
**Official Blind Review #2**

**Rating:** 6

**Review:**

This paper explores the use of replicating neurons across and within layers to compress fully connected neural networks. The idea is simple, and is evaluated on a number of datasets and compared with fully connected, single layer, and several compression schemes.

Strengths: a lot of nice experiments with clearly advantageous results are given.

Weaknesses: One obvious baseline missing is sparse compression, which can be achieved using either l1 regularization, or hard thresholding + fine tuning, both of which are easy to implement and appear in several works, e.g.

Scalable Neural Network Compression and Pruning Using Hard Clustering and L1 Regularization (Yang, Ruozzi, Gogate)
Training skinny deep neural networks with iterative hard thresholding methods (Yin, Yuan, Feng, Yan)

... many others just via googling ...

Also, I think this work should be compared with compression schemes that work via kronecker product, which seem very similar to this scheme (but where the kronecker matrix is binary to produce replication)

Compression of Fully-Connected Layer in Neural Network by Kronecker Product (Zhou, Wu)
(more via google)

One obvious advantage of replication over kronecker product is lower complexity, but nonetheless, the methods belong in a similar family.

Otherwise, I think the work makes sense, the idea is nice, and the results show promise!

After rebuttal: I have read the rebuttal and the authors have basically addressed all my concerns. It is a bit disappointing that simple L1 regularization can give competitive results, but the fact that the authors are willing to do the experiment and incorporate the results convinces me that there's nothing being hidden here, and the reader can make a fair and informed conclusion, so I have no more complaints.

**Experience Assessment:**

I do not know much about this area.

**Review Assessment: Checking Correctness Of Derivations And Theory:**

I did not assess the derivations or theory.

**Review Assessment: Checking Correctness Of Experiments:**

I assessed the sensibility of the experiments.

**Review Assessment: Thoroughness In Paper Reading:**

I read the paper thoroughly.

---

> ### Author Response · Authors · 2019-11-14
> **Answers to points brought up in review**
>
> Thank you very much for the feedback and the positive evaluation of our paper.
>
> > „One obvious baseline missing is sparse compression...“
>
> 1) Regarding the sparse compression baseline we want to point out, that the Bayesian Compression baseline [1] in our paper is implicitly sparsifying the network. Furthermore the authors compare their method against the sparsifying variational dropout proposed by [2] and show that they achieve better results.
> We performed some additional experiments employing simple L1 regularization and  L1 regularization combined with iterative hard thresholding (cp.  [3]) but without explicit cardinality constraint [4]. The results show that both methods in general perform a bit worse than the small FC baseline, beating our ACN in some of the cases where the small FC baseline is also stronger, especially for the two last parameter bins (with the highest number of parameters). However it doesn‘t change the overall impression and results. We will add the additional results to the appendix to clarify the points made.
>
> 		SparseL1	SparseL1+HT
> 		Accuracy	Accuracy
> har
> < 500	&	0.000	&	0.000	\\
> < 1000	&	0.181	&	0.193	\\
> < 2500	&	0.181	&	0.959	\\
> < 5000	&	0.981	&	0.967	\\
> >= 5000&	0.981	&	0.975	\\
> nomao
> < 250	&	0.000	&	0.000	\\
> < 500	&	0.000	&	0.000	\\
> < 1000	&	0.718	&	0.718	\\
> < 2500	&	0.952	&	0.949	\\
> >= 2500&	0.952	&	0.951	\\
> internetAds
> < 1000	&	0.916	&	0.913	\\
> < 2500	&	0.916	&	0.966	\\
> < 5000	&	0.969	&	0.966	\\
> < 10000&	0.977	&	0.966	\\
> >= 10000&	0.977	&	0.966	\\
> isolet
> < 2500	&	0.033	&	0.488	\\
> < 5000	&	0.033	&	0.933	\\
> < 7500	&	0.938	&	0.933	\\
> < 10000&	0.938	&	0.953	\\
> >= 10000&	0.938	&	0.953	\\
> spambase
> < 250	&	0.000	&	0.000	\\
> < 500	&	0.000	&	0.000	\\
> < 1000	&	0.733	&	0.739	\\
> >= 1000&	0.919	&	0.922	\\
> splice
> < 500	&	0.000	&	0.000	\\
> < 1000	&	0.533	&	0.533	\\
> < 2500	&	0.889	&	0.974	\\
> < 5000	&	0.971	&	0.974	\\
> >= 5000&	0.976	&	0.977	\\
> theorem
> < 250	&	0.000	&	0.000	\\
> < 500	&	0.000	&	0.000	\\
> < 1000	&	0.422	&	0.422	\\
> >= 1000&	0.488	&	0.493	\\
> bioresponse
> < 1000	&	0.548	&	0.548	\\
> < 2500	&	0.548	&	0.777	\\
> < 5000	&	0.767	&	0.791	\\
> < 10000&	0.788	&	0.791	\\
> >= 10000&	0.788	&	0.791	\\
> optdigits
> < 250	&	0.000	&	0.000	\\
> < 500	&	0.000	&	0.000	\\
> < 750	&	0.000	&	0.104	\\
> < 1000	&	0.104	&	0.961	\\
> >= 1000&	0.979	&	0.985	\\
>
> > „...this work should be compared with compression schemes that work via kronecker product, ...“
>
> 2) The proposed paper employing the Kronecker product is quite interesting. We will add it to the related work. Our 3rd baseline „TensorNet“ [5] (see section 4.2.2) employs the Tensor-Train (TT) format [6]. The TT format is itself a special case of a Nested Kronecker Tensor Decomposition [7].
> Furthermore [5] is well known in the network compression literature and comes with available code which simplifies the experiments. Therefore we argue that the TensorNet baseline is a good representation of compression methods based on layer-wise matrix decomposition and low-rank approximations. Furthermore our model does not only focus on layer-wise decompositions but takes the whole (deep) network structure into account.
>
>
>
> [1] Louizos, Christos, Karen Ullrich, and Max Welling. 2017. “Bayesian Compression for Deep Learning.” In Advances in Neural Information Processing Systems 30, edited by I. Guyon, U. V. Luxburg, S. Bengio, H. Wallach, R. Fergus, S. Vishwanathan, and R. Garnett, 3288–3298. Curran Associates, Inc. http://papers.nips.cc/paper/6921-bayesian-compression-for-deep-learning.pdf.
>
> [2] Molchanov, Dmitry, Arsenii Ashukha, and Dmitry Vetrov. 2017. “Variational Dropout Sparsifies Deep Neural Networks.” ArXiv:1701.05369 [Cs, Stat], June. http://arxiv.org/abs/1701.05369.
>
> [3] Han, Song, Jeff Pool, John Tran, and William Dally. 2015. “Learning Both Weights and Connections for Efficient Neural Network.” In Advances in Neural Information Processing Systems 28, edited by C. Cortes, N. D. Lawrence, D. D. Lee, M. Sugiyama, and R. Garnett, 1135–1143. Curran Associates, Inc. http://papers.nips.cc/paper/5784-learning-both-weights-and-connections-for-efficient-neural-network.pdf.
>
> [4] Jin, Xiaojie, Xiaotong Yuan, Jiashi Feng, and Shuicheng Yan. 2016. “Training Skinny Deep Neural Networks with Iterative Hard Thresholding Methods.” ArXiv:1607.05423 [Cs], July. http://arxiv.org/abs/1607.05423.
>
> [5] Novikov, Alexander, Dmitry Podoprikhin, Anton Osokin, and Dmitry Vetrov. 2015. “Tensorizing Neural Networks.” ArXiv:1509.06569 [Cs], September. http://arxiv.org/abs/1509.06569.
>
> [6] Oseledets, Ivan V. "Tensor-train decomposition." SIAM Journal on Scientific Computing 33, no. 5 (2011): 2295-2317.
>
> [7] Cichocki, Andrzej, Namgil Lee, Ivan V. Oseledets, A-H. Phan, Qibin Zhao, and D. Mandic. "Low-rank tensor networks for dimensionality reduction and large-scale optimization problems: Perspectives and challenges part 1." arXiv preprint arXiv:1609.00893 (2016).

---

### Official Review · AnonReviewer1 · 2019-10-20
**Official Blind Review #1**

**Rating:** 1

**Review:**

The paper describes a new method called Atomic Compression Network for constructing neural networks. The idea is straightforward. Basically, firstly create some neurons in random fashion, then reuse a subset of those neurons in each layer. The experiments shows ACN produces better accuracy than baseline models including a FC network, a Baysesina compression method, etc. for MINIST, etc. The paper also show ACN uses much less numbers of parameters and achieves similar accuracy when comparing with a large optima FC network on a set of datasets.

Overall, I don’t support accepting this paper. First, I don’t think the proposed idea is very innovative. Please elaborate why this method seems to work well when comparing baseline models. Is it just randomly constructed network also perform well?  Secondly, I’m not convinced we will use this method to build network in real world applications. The model size is small, but in what cases this small model size matters? Is this a reliable way to create useful models?

On page 7, in Figure 3, why logistic regression only has a single point in some of the plots?



**Experience Assessment:**

I have read many papers in this area.

**Review Assessment: Checking Correctness Of Derivations And Theory:**

I carefully checked the derivations and theory.

**Review Assessment: Checking Correctness Of Experiments:**

I carefully checked the experiments.

**Review Assessment: Thoroughness In Paper Reading:**

I read the paper at least twice and used my best judgement in assessing the paper.

---

> ### Author Response · Authors · 2019-11-14
> **Answers to clarification questions**
>
> Thank you for your time and the valuable feedback and insights regarding our paper.
>
> We would like to clarify some points mentioned in your review:
>
> > „Is it just randomly constructed network also perform well?“
>
> 1) No, naively randomly constructed networks do not perform well, as can be seen with the RER [1] baseline (section 4.2.2, figure 3 and table 5 in the appendix). The proposed method works well compared to the other baselines because the special weight sharing architecture enables the model to use the available capacity given by the number of its parameters more efficiently. As we show in the experimental section this applies to randomly constructed networks, where the shared weights are trained end-to-end what leads to an effective fine-tuned collective network. However we want to emphasize that only the distribution and connections of the neurons are random, while the number of layers and number of neurons per layer is predefined. Furthermore as we point out in the conclusion, the proposed method could be combined with smarter approaches to construct even more powerful networks, e.g. by using  NAS methods (cp. [7]).
>
>
> > „The model size is small, but in what cases this small model size matters?“
>
> 2) The small model size achieved by the presented methods matters in different theoretical and real world scenarios. An increasing number of recent publications are concerned with network compression approaches to improve scalability and minimize the required and utilized memory of originally huge models to run them on edge devices with restricted resources (IoT devices, smartphones, etc.) [2,3,4,5,6].
>
>
> > „Is this a reliable way to create useful models?“
>
> 3) The random construction of ACN is reliable and produces useful models, what is demonstrated by the reasonable variances shown in table 5 in the appendix. In the performed experiments on 9 diverse real world datasets with a different number of instances, features and classes as well as on 3 image datasets, the results show that the performance and gains of the proposed method are significant.
>
>
> > „On page 7, in Figure 3, why logistic regression only has a single point in some of the plots?“
>
> 4) Since logistic regression has a constant number of parameters and in figure 3 we compare models for different numbers of parameters, there can only be one point for logistic regression in all plots.
>
>
>
> [1] Cireşan, Dan C., Ueli Meier, Jonathan Masci, Luca M. Gambardella, and Jürgen Schmidhuber. "High-performance neural networks for visual object classification." arXiv preprint arXiv:1102.0183 (2011).
>
> [2] Cheng, Yu, Duo Wang, Pan Zhou, and Tao Zhang. 2017. “A Survey of Model Compression and Acceleration for Deep Neural Networks.” ArXiv:1710.09282 [Cs], October. http://arxiv.org/abs/1710.09282.
>
> [3] Kim, Yong-Deok, Eunhyeok Park, Sungjoo Yoo, Taelim Choi, Lu Yang, and Dongjun Shin. 2015. “Compression of Deep Convolutional Neural Networks for Fast and Low Power Mobile Applications.” ArXiv:1511.06530 [Cs], November. http://arxiv.org/abs/1511.06530.
>
> [4] Han, S., X. Liu, H. Mao, J. Pu, A. Pedram, M. A. Horowitz, and W. J. Dally. 2016. “EIE: Efficient Inference Engine on Compressed Deep Neural Network.” In 2016 ACM/IEEE 43rd Annual International Symposium on Computer Architecture (ISCA), 243–54. https://doi.org/10.1109/ISCA.2016.30.
>
> [5] Samie, Farzad, Vasileios Tsoutsouras, Lars Bauer, Sotirios Xydis, Dimitrios Soudris, and Jörg Henkel. 2016. “Computation Offloading and Resource Allocation for Low-Power IoT Edge Devices.” In 2016 IEEE 3rd World Forum on Internet of Things (WF-IoT), 7–12. https://doi.org/10.1109/WF-IoT.2016.7845499.
>
> [6] Mehta, Sachin, Mohammad Rastegari, Anat Caspi, Linda Shapiro, and Hannaneh Hajishirzi. 2018. “ESPNet: Efficient Spatial Pyramid of Dilated Convolutions for Semantic Segmentation.” In Computer Vision – ECCV 2018, edited by Vittorio Ferrari, Martial Hebert, Cristian Sminchisescu, and Yair Weiss, 11214:561–80. Cham: Springer International Publishing. https://doi.org/10.1007/978-3-030-01249-6_34.
>
> [7] Elsken, Thomas, Jan Hendrik Metzen, and Frank Hutter. 2018. “Neural Architecture Search: A Survey.” ArXiv:1808.05377 [Cs, Stat], August. http://arxiv.org/abs/1808.05377.

---

### Official Review · AnonReviewer3 · 2019-10-24
**Official Blind Review #3**

**Rating:** 1

**Review:**

This paper proposes a new way to create compact neural net, named Atomic Compression Networks (ACN). An immediate related work is LayerNet, where a deep neural net is created by replicating the same layer. Here, this paper extends replication down to the neuron level.

I am leaning towards rejecting this paper because the experimental setup is not well justified and a few important details are missing before conclusions can be drawn. I would like to ask a few clarification questions. Depending on the authors’ answers, I might be willing to adjust my rating.

(1) Is there missing a delta in the first half of line 6 in Algorithm 1?

(2) Throughout the experiments, for the same hyperparameter (e.g. Table 4 in A.2) do you run Algorithm 1 more than once and select the best sample architecture? If the answer is yes, summarizing all masks as one parameter will not be reasonable. Given a yes answer, I would also like to ask if the same number of samples have been considered for FC (for the same hyperparameter).

(3) Is there any intuition behind why FC does a much worse job of fitting curves than ACN with much less parameters? This refers to Fig. 2, if we compare FC with 41 parameters to ACN with 18 parameters. I am confused because MSE on sampled points often goes down when we increase the number of parameters for the application of curve fitting.

(4) Convolution can be thought of as a special case of ACN. ConvNet is the default architecture for working on image datasets. Since MNIST and CIFAR are considered, why not also compare to ConvNet?

(5) The claims that “ACNs achieve compression rates of up to three orders of magnitudes compared to fine-tuned fully-connected neural networks with only a fractional deterioration of classification accuracy” is quite misleading. Given fully-connected neural networks achieve up to 528 times with also a fractional deterioration (Sec. 4.3), by presumably having a shallower architecture.

**Experience Assessment:**

I have read many papers in this area.

**Review Assessment: Checking Correctness Of Derivations And Theory:**

I carefully checked the derivations and theory.

**Review Assessment: Checking Correctness Of Experiments:**

I carefully checked the experiments.

**Review Assessment: Thoroughness In Paper Reading:**

I read the paper thoroughly.

---

> ### Author Response · Authors · 2019-11-14
> **Answers to clarification questions**
>
> Thank you for your time and the thorough evaluation of our paper.
>
> In the following we want to clarify the points brought up in your review:
>
> 1) There is no delta missing. Line 6 in Algorithm 1 is meant as 2 consecutive lines since we return the two sets m and delta at the end of the alogirthm. We will separate it to make it more clear.
> However as you suggest it would also be possible to absorb the mask delta into m_i and only return m.
>
> 2) No, we do not select the best sample architecture. In the experiments we run Algorithm 1 only once per seed for 3 seeds and average the performance of all 3 resulting architectures. We do not apply any selection regarding the sampled architectures. All other models are also initialized, trained and evaluated over 3 different seeds and the results are averaged as well.
>
> 3) To motivate the benefit of recursively repeating neurons, we would like to present an example from function composition. Let us consider a simple function $f(x) = (\alpha x +\beta)^2, f: \mathbb{R} \rightarrow \mathbb{R}$ which has only two parameters $\alpha, \beta$. By applying the composition $f(f(x)) = \left(  \alpha^3 x^2 + 2 \alpha^2 \beta x + \alpha \beta^2 + \beta \right)^2$ we get a more complex function, but still having just two parameters $\alpha, \beta$. We can keep composing $f(\dots f( \dots f(x)))$ and achieve a very complex function, yet with only two parameters. Notice that the intuition of repeating neurons is equivalent to that of achieving a higher non-linear expressivity by composing functions, for instance composing a set of functions $f(x), g(x), h(x), \dots$ yields very deep representations, e.g. $f(g(f(h(g(h(f(x)))))$. Please consider that each $f, g, h$ can be a neuron, therefore our atomic networks are special cases of recursive function compositions from a set of base functions (a.k.a. repeating neurons in our paper). In our assessment, we are the first to consider adding non-linear expressivity by recursively applying the same set of neurons (a.k.a. functions).
>
> Therefore ACN achieves much deeper architectures with the same number of parameters compared to a standard FCN, what could further improve the fitting capability. Finally we see the expected trend that the fit for both models increases respectively when increasing the number of parameters when going from left to right in both rows of figure 2.
>
> 4) The main focus of this work is showing the advantage of ACN compared to MLP baselines on vector data. The image datasets were added for experimental diversity as special case of high dimensional vector data (the images were flattened) with an explicit structure. In the same way as MLPs, ACNs are not able to levarage the spatial information in image data compared to a specialized architecture like ConvNets. Furthermore, although ConvNets share parameters of their filters over the image, they do not share parameters between layers. Since the extension of the underlying idea of our ACNs to ConvNets is not feasible within the short time of the rebuttal periode, we plan to explore that direction in future work.
>
> 5) In our experiments we follow the established trend of comparing the compression rate w.r.t. a large standard model. However contrary to most work we also introduce a small, tuned FCN of comparable size to compressed networks, which is shown to be a very strong baseline [2].
> In general our experiments confirm the findings of [1], that with a very large and comprehensive hyperparameter search including the general network architecture, one can find very shallow and small FC networks which perform on par or even better than most networks produced by compression techniques. The inherent advantage of the compression techniques is however that in most cases they lead more reliably and with less computational effort to relatively small and well performing architectures.
> The 528 times is a typo in the text, it should be 218 times as reported in table 2. Furthermore the compression rates are achieved on different datasets e.g. the 1115 times of ACN compared to 133 of small FC on the internetAds dataset.
>
> [1] Liu, Zhuang, Mingjie Sun, Tinghui Zhou, Gao Huang, and Trevor Darrell. 2018. “Rethinking the Value of Network Pruning.” ArXiv:1810.05270 [Cs, Stat], October. http://arxiv.org/abs/1810.05270.
>
> [2] Chen, Wenlin, James Wilson, Stephen Tyree, Kilian Weinberger, and Yixin Chen. "Compressing neural networks with the hashing trick." In International Conference on Machine Learning, pp. 2285-2294. 2015.

---

### Author Response · Authors · 2019-11-15
**Updated Version for Rebuttal**

Many thanks to all the reviewers for providing valuable feedback and insights regarding our work.

Consequently we updated our paper to clarify the following key points in the review:

> Reviewer 2 rightfully proposed the additional comparison to sparsification methods employing L1-regularization and hard-thresholding. We added the respective experiment results as table 6 in the appendix. Furthermore we added the paper regarding compression with the Kronecker product to our related work.

> To empasize the importance of model compression which Reviewer 1 legitimately questioned, we added more respective references to our motivation in the introduction.

> In response to the points correctly brought up by Reviewer 3  we more extensively elaborate on the intuition behind the general model idea and the observed effects and competitive results. Furthermore we clarify the points regarding algorithm 1.

Besides the aforementioned points, we answered the raised questions in more detail in the direct comments to the reviews.

---

### Decision · Program_Chairs · 2019-12-19

**Decision:**

Reject

**Comment:**

This paper proposed a very general idea called Atomic Compression Networks (ACNs) to construct neural networks. The idea looks simple and effective.  However, the reason why it works is not well explained.  The experiments are not sufficient enough to convince the reviewers.